# Malignant Solitary Fibrous Tumours of the Pleura Are Not All the Same: Analysis of Long-Term Outcomes and Evaluation of Risk Stratification Models in a Large Single-Centre Series [note 1]

**DOI:** 10.3390/jcm12030966

**Published:** 2023-01-27

**Authors:** Sara Ricciardi, Delia Giovanniello, Luigi Carbone, Francesco Carleo, Marco Di Martino, Massimo Osvaldo Jaus, Sara Mantovani, Stefano Treggiari, Andrea Tornese, Giuseppe Cardillo

**Affiliations:** 1Unit of Thoracic Surgery, Azienda Ospedaliera San Camillo-Forlanini, Carlo Forlanini Hospital, 00151 Rome, Italy; 2PhD Program, Alma Mater Studiorum, University of Bologna, 40126 Bologna, Italy; 3Department of Cardio-Thoraco-Vascular Surgery, Sapienza University of Rome, Piazzale Aldo Moro 5, 00185 Rome, Italy; 4Unit of Anatomy and Pathological Histology, Azienda Ospedaliera San Camillo-Forlanini, Carlo Forlanini Hospital, 00151 Rome, Italy; 5Unicamillus—Saint Camillus University of Health Sciences, 00131 Rome, Italy

**Keywords:** solitary fibrous tumours, malignant pleural tumours, rare neoplasm, scoring system, surgery, outcomes

## Abstract

**Introduction:** Malignant solitary fibrous tumours of the pleura (mSFTP) are extremely rare diseases (<5% of all pleural neoplasms) with unpredictable behaviour. Surgery remains the standard of care for these tumours; however, estimating patient prognosis and planning follow-up remain challenging. Several risk stratification models have been proposed, but a classification with diagnostic and prognostic potential has not been well standardised yet. The aim of this study was to analyse the clinicopathological data of mSFTP to investigate their prognostic features and to compare the performance of three risk stratification models proposed in the literature. **Methods:** Observational retrospective cohort study on all proven cases of mSFTP surgically resected with radical intent between 2000 and 2019 in a single centre. Demographic, surgical and pathological data were examined. All patients were risk-stratified by using three prediction models: modified Demicco, De Perrot and Tapias. Overall survival (OS) and disease-free survival (DFS) were analysed. **Results:** There were 21 men and 13 women (median age, 67 years, range, 23–83 years). Twenty-one patients (62%) were symptomatic. The median follow-up was 111 months (range, 6–258 months). The 5-year OS and DFS were 81.2% and 77.4%, respectively. Nine patients (26.5%) experimented recurrences. At univariate analysis, the presence of necrosis (*p* = 0.019), nuclear atypia (*p* = 0.006), dimension greater than 11.5 cm (median value of our cohort) (*p* = 0.037) and relapse/disease progression (*p* = 0.001) were independent prognostic factor of worse OS. The administration of adjuvant treatment was a protective independent factor for survival (*p* = 0.001). Radicality of resection (*p* = 0.005); tumour dimension (*p* = 0.013), presence of necrosis (*p* = 0.041) and nuclear atypia (*p* = 0.007) and pleural pattern (*p* = 0.011) were independent prognostic factors of worse DFS. Analysing the three risk stratification models, the Tapias score was revealed as the best index to predict both OS (*p* = 0.002) and DFS (*p* = 0.047) in patients with mSFTP. **Conclusions:** Using the risk stratification model proposed by Tapias, patients with the highest risk of recurrence could be identified at the time of surgery to establish a more frequent imaging surveillance and longer follow-up. The role of adjuvant treatment in mSFTP therapy has not been established yet, but further analysis on patients with a high risk of recurrence, stratified according to risk models, along with biomolecular panels may tailor future post-surgical therapies.

## 1. Introduction

Solitary fibrous tumours are rare slow-growing mesenchymal neoplasms that can occur in many anatomical sites, most commonly the pleura [1].

During the years, they have been named in different ways, according to their origin and clinical course. Changes in their nomenclature have resulted in nothing but add more confusion to the description of their already difficult clinical and pathological patterns. Once their origin was established in the mesenchymal layer, a clearer path was outlined, setting the term “solitary fibrous tumours of the pleura” (SFTP) [2,3].

They can occur at any age, but most frequently in the sixth and seventh decades, and mostly present a favourable prognosis [4].

In fact, less than 5% of all pleural tumours have a malignant behaviour. It is important to assess their pathological state in order to prevent an adverse outcome, such as the development of local recurrence or metastatic disease [5].

According to the World Health Organization Classification of Tumours of the Pleura published in 2020, SFTP have been considered distinctive variants of existing tumour types, along with myxoid dermatofibrosarcoma protuberans (DFSP), DFSP with myoid differentiation, plaque-like DFSP, fat-forming (lipomatous), giant cell-rich SFT, epithelioid inflammatory myofibroblastic sarcoma and epithelioid myxofibrosarcoma. SFTP have been classified into three main groups: benign, which include the intermediate/locally aggressive category, NOS (not otherwise specified), which include the intermediate/rarely metastasizing category, and malignant [6,7]. A relatively small number of SFTs displays aggressive behavior with local recurrence, malignant transformation and distant metastasis, specifically, 10% to 30% [8].

SFTP have the potential to be difficult to treat and to predict. For this reason, during the years, there have been many attempts to understand the natural history of this disease.

Risk stratification models have been introduced in order to prevent and anticipate the possibility of recurrence. The principal characteristics of the proposed risk models are summarized in Table 1.

De Perrot et al. created a classification system based on four stages, indicating the growth pattern of the tumour (pedunculated or sessile) and the presence of malignancy, indicated by the presence of high cellularity with crowding and overlapping of the nuclei, cellular pleomorphism, high mitotic count (more than 4 per 10 high-power fields), necrosis or stromal/vascular invasion [9].

Tapias et al. developed a risk stratification model based on a combination of common clinical and histologic features of resected SFTPs, including pleural origin, morphology, size, hypercellularity, presence of necrosis or haemorrhage and number of mitoses, dividing the patients in groups at low and high risk of developing recurrences [10].

Demicco developed in its revised form a stratification model based on age, tumour dimension, number of mitoses and presence of necrosis (added later on in the revised form), thus dividing the patients in three main groups, at low, moderate and high risk of developing metastasis [11].

In our study, these risk stratification models were compared in terms of OS and DFS, after clinicopathological data analysis. 

## 2. Materials and Methods

The clinical medical records of all patients with malignant solitary fibrous tumours of the pleura, who were referred from January 2000 to December 2019 to our department (Unit of Thoracic Surgery, San Camillo Forlanini Hospital, Rome, Italy) were prospectively collected and retrospectively reviewed. All patients underwent surgery with radical intent. Patients <18 years of age, affect by sarcoma or other malignancies were excluded.

The diagnosis of mSFTP was obtained by imaging modalities (ultrasound and/or computed tomography) and percutaneous or endobronchial biopsy.

The malignancy criteria were stated according to England’s criteria, summarized in Table 2 [12].

Details concerning baseline demographic features, clinical presentations, comorbidities and pathological features were collected.

All patients had a cardiac and respiratory assessment before surgery.

The primary aim of the study was to analyse the clinicopathological data of mSFTP to investigate their prognostic features and to compare the performance of three risk stratification models proposed in the literature, based on histopathological and macroscopic features. The patients were retrospectively restaged according to following three prediction models: modified Demicco, De Perrot and Tapias. Overall survival (OS) and disease-free survival (DFS) were analysed. 

All patients were followed up at 1, 3 and 6 months after surgery with a clinical interview and a chest radiography/CT scan. Referral to medical or radiation oncologist was decided case by case in a multidisciplinary tumour board setting.

All patients signed an informed consent for the surgery and for the inclusion of their clinical data in the database. This study is a retrospective analysis of standard surgical procedures and was conducted in accordance with the Declaration of Helsinki. Institutional review board approval was not necessary, as stated by local law.

### Statistical Analysis

Baseline variables are described with percentages for categorical ones and mean and standard deviation (SD) for continuous ones.

Statistical analysis of surgical outcomes (operative time, redo surgery, hospital stay, complications) was performed. Statistical analysis of the long-term outcomes OS and DFS was performed. The Kaplan–Meier method was used to estimate the cumulative survival. COX proportional hazard regression was used for univariate and multivariate analyses. We considered values of *p* < 0.05 as statistically significant.

Overall survival was defined as the time from treatment to death, regardless of disease recurrence. Disease-free survival was defined as the time from diagnosis to the recurrence of the tumour or death.

Statistical analysis was conducted using SPSS statistic software (IBM SPSS statistics20 IBM Corporation, Chicago, IL, USA).

## 3. Results

The study included 21 men and 13 women who were histologically diagnosed with malignant SFTP according to England’s criteria. The median age was 67, ranging from 23 to 83 years.

Twenty-one patients (62%) were symptomatic: eight (23.5%) had dry cough, six (17.6%) experienced chest pain, six (17.6%) suffered from dyspnoea, and one (2.9%) had persistent fever with pleural effusion.

As it is described in Table 3, the great majority of the patients underwent exeresis of the tumour or wedge resection, 14 (41.2%) and 12 (35.3%), respectively. The preferred surgical approach was open surgery (31 patients, 91.2%).

Surgical radicality was obtained in most of the cases (91.2%), yet R1 was found in two patients (5.9%), and R2 in one patient (2.9%).

Data regarding the tumour characteristics were collected and later applied to the risk stratification models. The median diameter of the lesions was 11.5 cm (range, 5–29 cm).

SFTP originated mainly from the visceral pleura (82.4%) rather than from the parietal pleura (11.8%). Two patients showed inverted (intrapulmonary) SFTP (5.9%).

The majority presented a non-pedunculated growth pattern (67.6%). Twenty-three patients (67.6%) showed a mitotic count/10 HPF (High-Power Fields) greater than 4. In twenty-eight cases (82.4%), necrosis or haemorrhagic areas were present. Hypercellularity was shown in twenty-three patients (67.6%).

Bcl2 positivity was found in 26 patients (76.4%), CD34 positivity in 28 patients (82.3%), and STAT6 positivity in 32 patients (94.1%) (Figure 1).

Vimentin and cytokeratin (AE1-3) were found in four (11.7%) and two (5.8%) patients respectively.

The patients were retrospectively restaged according to three prediction models: modified Demicco, De Perrot and Tapias, as showed in Table 4.

The median follow-up was 111 months, ranging from 12 to 258 months. The 5-year OS was 81.2%, and the 5-year DFS was 77.4%. Recurrences were found in nine patients (26.5%).

In univariate analysis, the presence of necrosis (*p* = 0.019), nuclear atypia (*p* = 0.006), dimension greater than 11.5 cm (median value of our cohort) (*p* = 0.037), high Ki67 expression (*p* = 0.018) and relapse/disease progression (*p* = 0.001) were associated with a worse OS. The administration of adjuvant treatment was a protective factor for survival (*p* = 0.001). The Kaplan–Meier curves for OS according to the tumours’ characteristics are reported in Figure 2.

In multivariate analysis, adjuvant therapy and relapse were confirmed prognosticators (Table 5).

Radicality of resection (*p* = 0.005), tumour dimension (*p* = 0.013), presence of necrosis (*p* = 0.041), nuclear atypia (*p* = 0.007), high expression of Ki67 (*p* = 0.012) and pleural pattern (*p* = 0.011) were associated with worse DFS. In multivariate analysis, only surgical radicality was confirmed as an independent prognostic factor of worse DFS (Table 6).

When analysing the three risk stratification models, the Tapias score (low versus high score) was revealed as the best index to predict both OS (*p* = 0.002) and DFS (*p* = 0.047) in patients with mSFTP. (Table 7 and Figure 3). The capacity of the de Perrot stage (stage 2 versus 3) to predict OS and DFS did not reach significance, whereas the modified Demicco score (low and moderate versus high) was closer to statistical significance to predict DFS.

## 4. Discussion

SFTP are rare neoplasms arising from the mesenchymal layer. Due to their possible evolution to malignant tumours, they have been challenging to treat and to predict. No data are available regarding risk factors that may lead to their growth, nor environmental factors promoting their development are known [13,14].

These tumours are most frequently pleura-based but can also grow into the lung parenchyma, without prevalence in the right or left side [14,15].

Their clinical behavior varies widely, with cases being mostly asymptomatic, and the tumour discovered as an incidental finding. If present, the most common symptoms are chest pain, dyspnoea and cough [1,3,14,15]. They can arise due to irritation of the adjacent areas, pleural effusion formation and paracrine action of biochemical substances released by the tumour [3].

Some studies state that if present, these symptoms may lead to a higher chance of malignancy development [16].

When unusual symptoms such as hypoglycaemia or articular rigidity arise, it is important to never underestimate their presence, since they could be a potential red flag for paraneoplastic syndromes.

Hypoinsulinemic hypoglycaemia characterizes Doege Potter Syndrome (DPS), which leads to an excessive ectopic secretion of insulin-like growth factor 2 (IGF2) by the tumour. Laboratory blood analyses show increased IGF2 levels and low insulin and peptide C levels [17,18].

Those patients who present with bilateral articular pain, swelling, rigidity, joint stiffness and digital clubbing are most likely to be affected by Pierre–Marie–Bamberger syndrome (PMBS), a rare syndrome yet more common than DPS [19,20].

In our study, twenty-one patients (62%) were symptomatic. However, the presence of symptoms seemed not to be related to OS or DFS. It is highly crucial to make the right differential diagnosis for SFTP and all those masses that share common features in terms of anatomical, histopathological and clinical findings.

More precisely, intrapulmonary fibrous tumours must be differentiated, especially, from squamous cell carcinomas, sarcomas, carcinoids and malignant mesotheliomas, for the right clinical and therapeutical choice [21].

For example, Leucotere et al. presented three cases of a morphological mSFTP with unexpected strong cytokeratin expression, which is usually present in sarcomas and mesotheliomas [22].

The macroscopic appearance of these tumours is highly connected to their malignancy index, as large dimensions lead to a higher chance of tumoral cells presence [23].

In our study, the median maximum diameter of the lesions was 11.5 cm (range 5–29 cm), confirming this general agreement. Indeed, a tumour dimension >11.5 cm appeared as an independent prognostic factor of worse OS and DSF.

The histological presentation of these tumours lays the foundation for a possible prognostic risk model stratification. Macroscopic features include a well circumscribed, almost completely encapsulated tumour, hard in consistency, with a clean, whitish cut surface, necrotic areas within and sometimes a vorticoid appearance, alternating with myxoid areas. The microscopic appearance is typically characterized by a “patternless pattern”, with the proliferation of elongated or ovoid cells with varying amounts of connective tissue, alternating patterns of fibrosis, hypercellularity and hypocellularity, a collagenous stroma, and typical “staghorn”-shaped (hemangiopericytoma-like or HPC-like) blood vessels [21].

In our series, necrosis or haemorrhagic areas were present in 28 patients (82.4%), and 23 patients (67.6%) showed a mitotic count/10 HPF greater than 4. Hypercellularity was shown in twenty-three patients (67.6%). Necrosis and nuclear atypia were confirmed as independent prognostic factors of both OS and DFS in our study.

A strong diagnostic impact is provided by immunohistochemistry. SFTP are pathophysiologically characterized by a translocation leading to the fusion of the NGFI-binding protein 2 (NAB2) gene to the transcription activator gene 6 (STAT6). The expression of CD34, CD99 and BCL-2 genes is common in SFTP, yet not specific. For example, CD34 is found in only 5–10% of SFTP, mainly in malignant and dedifferentiated tumours [24,25].

Interestingly, a high proportion of our cases expressed CD34 (82.3%). Moreover, Bcl2 positivity was detected in 76.4% and STAT6 positivity in 94.1% of our cohort.

Cytokeratins, ALDH1 and GRIA2 are markers that are usually expressed as well, but have a fluctuating behavior [26].

Even the negativity of the expression of some genes such as EMA, H-caldesmon, desmin, CD31, S100 is used for diagnosis, especially to differentiate SFTP from other soft tissue tumours [3,21].

The presence of vimentin and cytokeratin (AE1-3) was also found in 11.7% and 5.8% of the patients, respectively.

Ki67 is a well-known aggressiveness index and, if its expression results to be 10% or greater, it is significantly associated with tumour recurrence [27]. A high expression of Ki67 was confirmed as an independent prognostic factor of both OS and DFS in our cohort.

Whether by VATS or open, the aim of surgery is to obtain complete resection margins. If R0 is not achieved, the surgical intervention cannot be considered a radical treatment. Sometimes, even if R0 is obtained, recurrence is possible in patients with both malignant and benign variants [28]. Although in most cases a tumour excision, possibly associated with a minor lung parenchymal resection, was required, two patients (5.9%) of our cohort were treated by pneumonectomy. This approach was carefully evaluated and was considered necessary due to the size and central location of their tumours.

Complete resection margins were achieved in 31 patients (91.2%) in our series. Radicality of resection resulted to be an independent prognostic factor of worse DFS (*p* = 0.005), but not of OS (*p* = 0.912), as expected, since OS does not include recurrence in its timeframe.

Based on the above-mentioned characteristics, prognostic risk models aim to predict the pathological behaviour of SFTP.

Analysing the three risk stratification models applied to our cohort, we believe that the Tapias score is the best index to predict both OS and DFS in patients with mSFTP.

## 5. Conclusions

SFTPs are prognostically challenging tumours with unpredictable behavior. The cornerstone of mSFTP treatment remains radical surgery, with negative resection margins as the most important prognostic factor. At present, a global consensus on the treatment of mSFTPs is missing, and randomized clinical trials are mandatory to define the role of systemic therapies for these rare disease entities. A multidisciplinary tumour board assessment is also crucial to define the correct management of these malignancies.

Our findings suggest that using the risk stratification model proposed by Tapias, patients with the highest risk of recurrence could be identified at the time of surgery to establish a more aggressive treatment strategy, frequent imaging surveillance and longer follow-up.

The role of adjuvant treatment for mSFTP has not been established yet, but further analysis on patients with a high risk of recurrence, stratified according to risk models, along with biomolecular panels may tailor future post-surgical therapies.

## 6. Limitation

The most important limitation of the present study is its retrospective nature, with a small number of included cases. Moreover, the limited number of recurrence and metastases restricts the significance of the multivariate analysis for DFS; yet our results match those of other large datasets on mSFPT.

mSFTP are rare tumours; thus, the limited number of patients statistically reduces the chance of obtaining solid results. However, due to the homogeneity of our cohort and its direct relation to the intrinsic characteristics of the tumour and the absence of similar large groups previously analysed in the literature, the outcomes of this study can be considered reliable.

## Figures and Tables

**Figure 1 jcm-12-00966-f001:**
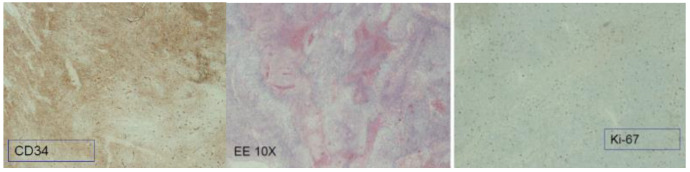
Immunoistochemistry analysis of mSFTP. Coloration for CD34, Ki-67 and Hematoxylin eosin 10X.

**Figure 2 jcm-12-00966-f002:**
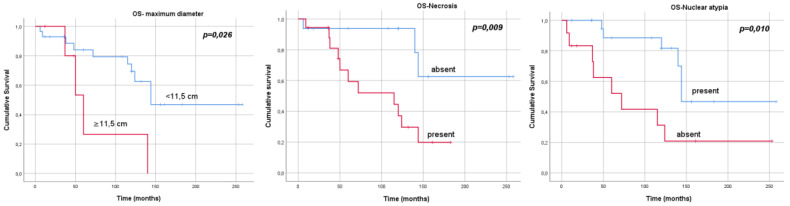
Kaplan–Meier curves, overall survival and tumours’ characteristics.

**Figure 3 jcm-12-00966-f003:**
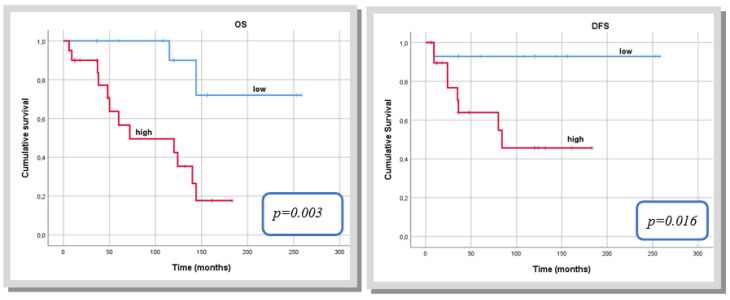
Kaplan–Meier curve for overall survival and disease-free survival according to the Tapias score.

**Table 1 jcm-12-00966-t001:** Criteria and scoring systems of the three different stratification models.

Tapias		Modified Demicco		de Perrot
Criterion	Score	Criterion	Score	Criterion
Morphology Pedunculated Sessile	0 1	Age (years) <55 ≥55	0 1	
Pleural origin Visceral/intrapulmonary Parietal	0 1			Vascular/pleural invasion
Dimension (cm) <10 ≥10	0 1	Dimension (cm) <5 5–10 10–15 ≥15	0 1 2 3	Pleomorphism
Mitoses (/10 HPH) 0–3 ≥4	0 1	Mitoses (/10 HPF) 0 1–3 >4	0 1 2	Mitoses ≥4/10 HPF
Hypercellularity	1			Hypercellularity
Necrosis and/or haemorrhage	1	Necrosis Absent Present	0 1	Necrosis haemorrhage
Risk Of Recurrence Low High	Total 0–2 3–6	Risk Of Recurrence Low Moderate High	Total 0–3 4–5 6–7	Stages 0: pedunculated tumour without signs of malignancy 1: sessile/inverted tumour without signs of malignancy 2: pedunculated tumour with signs of malignancy 3: sessile/inverted tumour with signs of malignancy 4: metastatic or multifocal disease at presentation

**Table 2 jcm-12-00966-t002:** England’s criteria of malignancy [12].

Dimension >10 cm
Tumour necrosis or haemorrhage
Increased cellularity
Nuclear pleomorphism
>4 mitoses/10 HPFs

HPF: High-Power Field.

**Table 3 jcm-12-00966-t003:** Surgical procedures, tumour characteristics and risk stratification.

Surgical procedures Exeresis Wedge resection Lobectomy Pneumonectomy	14 (41.2%) 12 (35.3%) 6 (17.6%) 2 (5.9%)
Surgical approach: Open Minimally invasive	31 (91.2%) 3 (8.8%)
Radicality: R0 R1 R2	31 (91.2%) 2 (5.9%) 1 (2.9%)
Pleural pattern Parietal Visceral Inverted (intrapulmonary)	4 (11.8%) 28 (82.4%) 2 (5.9%)
Growth pattern Nonpedunculated Pedunculated	23 (67.6%) 11 (32.4%)
Mitosis >4/10 HPF	23 (67.6%)
Necrosis or haemorrhagic areas	28 (82.4%)
Hypercellularity	23 (67.6%)

**Table 4 jcm-12-00966-t004:** Risk stratification assessment.

De Perrot Stage 2 Stage 3 Stage 4	11 (32.4%) 23 (67.6%) 0
Modified Demicco Low Moderate High	10 (29.4) 9 (26.5%) 15 (44.1%)
Tapias Low High	14 (41.2%) 20 (58.8%)

**Table 5 jcm-12-00966-t005:** Univariate and multivariate analyses for OS.

Variable		Univariate Analysis		Multivariate Analysis	
		HR (95% CI)	*p*-Value	HR (95% CI)	*p*-Value
Age	<67 ≥67		0.199		
Gender	F M		0.445		
Symptoms	Absent Present		0.275		
Comorbidities	Absent Present		0.976		
**Maximum diameter**	**>11.5**	**3.712 (1.083–12.717)**	**0.037**		0.909
**Necrosis**	**Absent** **Present**	**4.649 (1.286–16.799)**	**0.019**		0.630
Haemorrhagic areas	Absent Present		0.287		
Calcification	Absent Present		0.328		
**Nuclear Atypia**	**Absent** **Present**	**3.856 (1.461–10.176)**	**0.006**		0.273
**Pleural pattern**	**Parietal** **Visceral** **Inverted**	**2.662 (1.275–5.559)**	**0.009**		0.524
**Ki67**	**>10%**	**3.609 (1.245–10.467)**	**0.018**		0.840
Approach	MIS (VATS/ROBOT)		0.197		
	Thoracotomy				
Type of surgery	Wedge		0.210		
	Exeresis				
	Segmentectomy				
	lobectomy				
	other				
**Adjuvant therapy**	Administered Not administered	**0.041 (0.007–0.252)**	**0.001**	**0.024 (0.002–0.269)**	**0.003**
Radicality	R0		0.912		
	R1				
	R2				
**Relapse**	**No** **yes**	**6.854 (2.253–20.849)**	**0.001**	**7.279 (1.511–35.058)**	**0.013**

**Table 6 jcm-12-00966-t006:** Univariate and multivariate analyses for the DFS.

Variable		Univariate Analysis		Multivariate Analysis	
		HR (95% CI)	*p*-Value	HR (95% CI)	*p*-Value
**Age**	<67 ≥67		0.975		
Gender	F M		0.641		
Symptoms	Absent Present		0.128		
Comorbidities	Absent Present		0.553		
**Maximum diameter**	**>11.5**	**5.990 (1.461–24.553)**	**0.013**		0.096
**Necrosis**	**Absent** **Present**	**8.901 (1.102–71.862)**	**0.041**		0.566
Haemorrhagic areas	Absent Present		0.052		
Calcification	Absent Present		0.468		
**Nuclear Atypia**	**Absent** **Present**	**6.860 (1.679–28.025)**	**0.007**		0.268
**Pleural pattern**	**Parietal** **Visceral** **Inverted**	**2.882 (1.281–6.486)**	**0.011**		0.978
**Ki67**	**>10%**	**5.299 (1.451–19.356)**	**0.012**		0.175
Approach	MIS (VATS/ ROBOT)		0.183		
	Thoracotomy				
Type of surgery	Wedge		0.612		
	Exeresis				
	Segmentectomy				
	lobectomy				
	other				
Adjuvant therapy	Administered Not administered		0.270		
**Radicality**	**R0**	**0.032 (0.003–0.356)**	**0.005**	**0.051 (0.004–0.585)**	**0.017**
	**R1**				
	**R2**				

**Table 7 jcm-12-00966-t007:** Univariate analysis for OS and DFS, risk stratification models.

	HR (95% CI)	*p*-Value
De Perrot OS DFS	0.592 (0.200–1.754) 0.641 (0.171–2.403)	0.344 0.512
Modified Demicco OS DFS	1.757 (1.878–3.519) 2.927 (1.965–8.884)	0.094 0.058
Tapias OS DFS	6.895 (1.533–31.015) 8.300 (1.029–66.948)	0.002 0.047

## Data Availability

The data underlying this article will be shared on reasonable request to the corresponding author.

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
