# Peer review of "Malignant Solitary Fibrous Tumours of the Pleura Are Not All the Same: Analysis of Long-Term Outcomes and Evaluation of Risk Stratification Models in a Large Single-Centre Series†"

_jcm, 2023, doi:10.3390/jcm12030966_

Round 1

Reviewer 1 Report

In this paper the Authors present a retrospective study on a cohort of 34 patients with malignant solitary fibrous tumour of the pleura (mSFTP) surgically treated in a single center over a period of twenty years.

The clinical characteristics and management of these patients were analysed and patients were risk-stratified by using three prediction models (modified Demicco, De Perrot and Tapias). Prognostic factors of worse overall survival (OS) and disease-free survival (DFS) were analysed. The Authors conclude that Tapias score revealed to be the best model to predict both OS and DFS in mSFTP.

The paper is relevant, dealing with a specific topic of interest, even if regarding a rare disease (mSFTP) and is very well written and documented by good tables and figures.

The novelty and strength of the study as well as its limitations are well addressed and discussed.

I have just one question and some suggestions.

1) In the Title “Malignant fibrous solitary tumour of the pleura etc…” should be corrected with “Malignant solitary fibrous tumour of the pleura etc…”

2) In the Abstract, in the “Conclusions” (page 1, line 44) please correct “may tailored” with “may tailor”.

3) In the “Materials and Methods” (page 3, line 112) please correct “bord” with “board”.

4) In the “Results”, in Table 1 (pages 3-4), concerning the surgical procedures performed, the Authors report two patients undergoing pneumonectomy, that is a large pulmonary resection, demanding for the patient: was it performed due to the dimension and to the location of the tumor at the hilum? Please specify these details making a comment in the Discussion to justify such a large resection.

4) In the “Results”, figure legend of Figure 1 is missing (page 4, lines 161-162).

5) In Table 3 (page 5) and Table 4 (page 7) are there any missing fields in the tables? Or are they correct that way, with some fields empty?

6) In the “Results”, concerning DFS prognosticator (Page 6, line 189) I suppose the reference is to table 4 and not 3.

7) In the “Discussion”, is the word “Cytocherines” correct? (page 9, line 265).

Author Response

We would like to thank the reviewer for the suggestions and the time dedicated to our study. 

We hope this version, with the required changes and improvements, is in line with what you requested.

Reviewer 2 Report

Comments for Authors

Dear Authors,

Thank you for the opportunity to review your manuscript, it was of interest to read about this under-represented topic that poses a challenge for clinicians in the field.

Please see below some comments and suggestions for edits, with page and line numbers:

Page 2

Line 68

Consider changing ‘small part’ to ‘small number’

Page 2

Line 71

Consider changing ‘the ability’ to ‘the potential’

Page 2

Line 72

Consider changing ‘reach a final pattern to follow’ to ‘understand the natural history of the disease’

Page 2

Helpful to include a table of the characteristics and scoring mechanisms for the different risk prediction/ prognostic models

Page 3

Line 100

Helpful to include a table of England’s criteria, or an abridged version if too lengthy

Page 3

Line 147 Define HPF (I think it means ‘High Powered Field’?)

Table 1

Spelling: Hypercellularity

Figure 1

Title not completely showing (formatting issue)

Page 4

Could you elaborate on the significance of the different proteins you’ve included for the immunohistochemistry analysis (Are these considered diagnostically important, is it contained within England’s criteria, if so another reason to include England’s criteria)

Page 5

Table 3

I think the wording around ‘Independently associated’ might cause some confusion when referring to your Univariate model.

In my understanding, Univariate analysis (or unadjusted) means the association found between the variable and the outcome of interest has not been adjusted for the effect other variables might have had on the outcome (ie confounders). Whilst in Multivariate (or adjusted) analysis, you identify the relationship of a variable to the outcome of interest, once other variables are taken into account (so a more representative and accurate description of the relationship, provided assumptions are correct).

Therefore, I would interpret this as: Adjuvant therapy and Relapse are the ‘Independently variables associated with survival, once accounting for confounders.’

Need to make clear what the reference category you are comparing against is to make sense of the HRs presented (Gender: Male or Female as the reference standard 1, Symptoms absence or presence as the reference standard 1, Comorbidities present or absent, etc)

Similar issues for Table 4

Radicality would be the only Independent associated variable with DFS

Table 5

Please input what was the reference range (De Perrot stage 2, Demico Low, Tapias Low as the reference standard against which the regression models were run, ie HR 1?)

Page 8

Line 212

Seems to contradict: pleural based but also belongs to lung parenchyma?

Page 8-9

A lot of this initial text in the Discussion would be better placed in the Introduction (the presentation of mSFTP, symptoms, paraneoplastic associations etc)

You should mainly just focus on the interpretation of your findings, presented in the Results section.

Page 9

What is ‘HPC-like’? (full text of abbreviation needed)

I would probably summarise in the Discussion, what the Independent risk factors for poor outcome were (for OS and DFS), and any clinical interpretation you might recommend on this basis (ie  you are *less worried* for patients who have undergone adjuvant therapy, or any who relapse have poor prognosis, so you might wish to identify relapse early – how do you propose this, surveillance imaging etc?, how do you guarantee radicality of resection margin?)

Some explanation for why Tapias is better – we need to see what parameters this model is based on to understand this finding better (again if all 3 models with their scoring systems are presented in the Introduction, it might be easier to interpret your findings from the study).

In the limitations, it might be helpful when recommending validity to show that compared to the other large datasets on mSFPT, rates of relapse/ metastases etc are comparable (and if not, any intrinsic differences in your population vs other studies).

Author Response

Many thanks for the time and efforts you have dedicated to our study. 

Your suggestion was highly appreciated. We hope this version is in line with what you requested.
